# Iron Deficiency Anemia in Celiac Disease

**DOI:** 10.3390/nu13051695

**Published:** 2021-05-17

**Authors:** Valentina Talarico, Laura Giancotti, Giuseppe Antonio Mazza, Roberto Miniero, Marco Bertini

**Affiliations:** 1Department of Pediatric, Pugliese-Ciaccio Hospital, 88100 Catanzaro, Italy; roberto.miniero@unicz.it; 2Unit of Pediatrics, University “Magna Graecia”, 88100 Catanzaro, Italy; lauragiancotti@virgilio.it; 3Department of Pediatric Cardiology, Regina Margherita Hospital, Città della Salute e della Scienza, 10126 Torino, Italy; giuseppeantoniom@gmail.com; 4R&D Department, Laboratori Baldacci SpA, 56121 Pisa, Italy; bertini@baldaccilab.com

**Keywords:** Iron deficiency Anemia, Celiac disease, iron absorption

## Abstract

The iron absorption process developsmainly in the proximal duodenum. This portion of the intestine is typically destroyed in celiac disease (CD), resulting in a reduction in absorption of iron and subsequent iron deficiency anemia (IDA). In fact, the most frequent extra-intestinal manifestation (EIM) of CD is IDA, with a prevalence between 12 and 82% (in relation with the various reports) in patients with new CD diagnosis. The primary treatment of CD is the gluten-free diet (GFD), which is associated with adequate management of IDA, if present. Iron replacement treatment historically has been based on oral products containing ferrous sulphate (FS). However, the absorption of FS is limited in patients with active CD and unpredictable in patients on a GFD. Furthermore, a poor tolerability of this kind of ferrous is particularly frequent in patients with CD or with other inflammatory bowel diseases. Normalization from anemic state typically occurs after at least 6 months of GFD, but the process can take up to 2 years for iron stores to replenish.

## 1. Introduction

Iron deficiency anemia (IDA) represents a frequent medical condition encountered in clinical practice by general practitioners, pediatricians and several other specialists. In Western countries, the prevalence of IDA is higher in two phases of the pediatric age: one occurs between the first and third year of life (2.3–15%) and the second occurs in adolescence (3.5–13% in males, 11–33% in females); in adults the prevalence is less than 1% in men <50 years of age, 2–4% in men >50 years of age, 9–20% in menstruating teenagers and young women, and 5 to 7% in post-menopausal women [1,2,3,4].

The most common pathogenetic mechanisms of IDA in adults are increased menstrual flow, occult intestinal bleeding or reduced iron absorption, as occurs in celiac disease (CD). In children inadequate intake, increased daily requirement and CD are the leading causes [1,2,3,4].

The typical Western diet contains up to 20 mg of daily iron intake, of which 1–2 mg are absorbed; 85–90% of iron is in the non-heme form, mainly as ferric iron (Fe^3+^), that has to be converted into the ferrous form (Fe^2+^) in order to be absorbed. This process is regulated by the cytochrome B (DCYTB), a ferrireductase located on the apical membrane of duodenal enterocytes. The absorption of Fe^2+^ primarily occurs in the proximal duodenum, at the brush border of the mucosa cells, through a membrane transport protein called divalent metal transporter (DMT1). Otherwise, heme-iron is absorbed in the same bowel district, but separately from DMT1 and more efficiently than inorganic iron [1,2,3]. This portion of the duodenum represents the most frequently destroyed region in CD, consequently resulting in a reduction in iron absorption, and therefore IDA, which is another of the most frequent extra-intestinal manifestations (EIM) of CD [5,6,7,8,9,10,11,12]. The prevalence of anemia, different due to the high variability of the studies on this topic, is estimated at between 12 and 82% in patients with new CD diagnosis and about 46% in patients affected by subclinical CD [5,6,7,8,9,10,11,12]. Furthermore, as noted by Berry, the finding of IDA in patients with CD is less frequent in works from the West (Europe and North America) compared to Asia, Africa and the Middle East; probably, this observation reflects the higher prevalence of IDA, independently from CD, in different regions. Studies focused on prevalence of IDA in celiac patients are summarized in Table 1 [13,14,15,16,17,18,19,20,21,22,23,24,25,26].

IDA often represents the only clinical sign of CD both in children and in adults, especially in patients with subclinical/atypical forms of CD [22,23,24]. Kocharet al. [22] reported that 39% out of 434 CD patients had anemia as the unique presenting feature. In a multicenter study, including 1026 Italian patients with subclinical/silent CD, the most frequent EIM was IDA (about 39%), found in 46% of adults and 35% of pediatric patients [17].

The counterpart of the relationship between CD and IDA is the high frequency of CD in IDA patients compared to non-anemic ones. However, these studies show a wide variability and diversity in diagnostic methods for CD, and are often performed in selected populations. In a recent systematic meta-analysis, Mahadev identified 18 studies including 2998 patients (adults and children) with IDA, finding CD in 3.2–5.5% of individuals [27]. Shahriari et al. studied the prevalence of IDA in 184 children (92 patients with IDA responsive to iron supplementation, 45 patients with IDA unresponsive to iron supplementation). The authors showed a positive serology of CD in 5.4% and in 28.3% of children with responsive and refractory IDA, respectively [5]. Ertekin et al. reported that among 61 children with IDA, 21.3% had positive serology for CD [28] and Karaman et al. found positive serology for CD in about 8.4% of 250 children with IDA [29]. Kalayciet al. reported a prevalence of CD in about 4.4% of children with IDA [30] and Abd El Dayemet al. in about 44% of 25 children with refractory IDA [31]. Other studies showed different important information: Bansal et al. diagnosed CD in 83 Indian children with IDA unresponsive to conventional therapy [32]. About adults, Dube et al. reported a prevalence of CD between 2.9% and 6% in patients with IDA that increased to 10–15% in patients with IDA and concomitant gastrointestinal symptoms [33]. Hershko, in a cohort of 325 patients, found a 5.2% prevalence of CD [34]. Similar data were obtained by other authors: Corrazza et al. (5.0% out of 200 patients) [35], Carroccio et al. (5.8% out of 85 patients) [24], Annibale (5.6% out of 71patients) [36], Howards (4.7% out of 258 patients) [37], Mandal (1.8% out of 504 patients) [38], Carter et al. (6.0% out of 116 patients) [39] and Lasa (11.11% out of 135 patients) [40].

Paez et al. studied the CD diagnosis delay in 101 adult patients, 52 with gastrointestinal symptoms and 49 without one. Anemia was found in 69.4% of patients without gastrointestinal symptoms, compared with 11.5% in the group with gastrointestinal symptoms. The results showed a mean diagnostic delay of approximately 2.3 months for the group of patients with gastrointestinal symptoms and of 42 months for the group without gastrointestinal manifestations [41]. 

Given the treatable nature of CD, it is of importance not to delay the diagnosis in order to reduce the effect of the disease on health of the affected subjects. Since patients with IDA, especially in the case of non-responsive forms to treatment, have a higher risk of having CD than the general population, screening with tissue transglutaminase antibodies is strongly recommended in these subjects. Several scientific societies such as the European Society for Paediatric Gastroenterology Hepatology and Nutrition (ESPGHAN) [42], the American Gastroenterological Association (AGA) [43], the British Society of Gastroenterology (BSG) [44] and the Italian Association of Hospital Gastroenterologist and Endoscopist (AIGO) and the Italian Society of Pediatric Gastroenterology Hepatology and Nutrition (SIGENP) have drawn up their own guidelines on this topic [45]. However, in clinical practice these recommendations remain often unfollowed. Smukalla found that hematologists rarely screen for celiac disease in the initial workup of IDA patients. According to these authors, this attitude could be a determinant of the underdiagnosis of CD in the United States [46].

After about 24 months of a GFD, an improvement in IDA can be observed [47,48]. De Falco, studying 505 adults with CD, of which 45% had IDA, showed a persistence of anemia after one year of GFD even in the presence of histological normalization of the duodenal mucosa [20].

Saukkonen et al., among 163 adult CD patients, showed greater severity of CD in anemic patients than in non-anemic ones (tTG 65 versus 26.4 U/mL), respectively. The authors showed that the villous height–crypt ratio was lower in anemic patients than in non-anemic ones, with an increased in intraepithelial lymphocytosis, after one year on GFD [49]. Various studies showed similar results; Abu Daya et al. showed a worse villous atrophy in patients with anemia than those presenting with only gastrointestinal symptoms at CD diagnosis [18]. Nurminen et al. demonstrated a greater degree of villous atrophy in patients with CD and IDA, both compared to those with gastrointestinal symptoms at the diagnosis of CD and those whose diagnosis of CD was made following a screening test for serum CD performed for familiarity) [50]. Considering that the presence of extraintestinal manifestations alone may be due to the delayed diagnosis of CD, this is likely to lead to worsening of both the anemia and the degree of villous atrophy at diagnosis. 

Two other large series subsequently confirmed the hypothesis of correlation between severity of anemia and severity of intestinal atrophy [15,51].

Similar results have been obtained in children. Repo et al. showed that the median values of hemoglobin, iron, ferritin and transferrin saturation were significantly lower in children with CD and total intestinal villous atrophy compared to those with partial/subtotal atrophy, with potential CD and children belonging to the control group [52].

Schieppattiet al. [53] demonstrated a significant correlation between an increase in hemoglobin concentration and adherence to GFD; Stefanelli et al. [54] and Annibale et al. [36] evidenced a correlation between improving hemoglobin levels and decrease of histological scores of duodenal lesions after starting of a GFD. A normalization of the hemoglobin value was demonstrated after about 6–12 months from the beginning of the GFD alone, following the restoration of the integrity of the intestinal mucosa. However, data showed that only 50% of patients showed normalization of iron deficiency after 12 months on a GFD [36].

There are several factors that could explain the reason for the persistence of IDA after the adoption of a GFD and oral iron supplementation: non-adherence to a GFD, the presence of ultrastructural and/or molecular alterations of the enterocytes despite the reformation of the integrity of the duodenal mucosa or the presence of specific genetic factors [54].

## 2. Pathogenesis

Repo et al., evaluating the iron transporter protein expressions in children with CD, found an increased expression of ferroportin and a decreased expression of hephestin in children with histologically confirmed celiac disease compared with the non-celiac controls. There were no other significant differences between the study groups in the expression of iron transporter proteins. In addition, no differences in any of these proteins were detected when it involved anemic and non-anemic children [55].

Several studies have investigated whether it is possible to identify a possible genetic predisposition to IDA in MC. Barisaniet al. showed that the expression of some iron regulatory proteins (DMT1, DCYTB, ferroportin 1, efestin and transferrin receptor) was similar in 25 patients with CD compared to 10 controls [56]. Similarly, Sharma et al. showed greater expression of DMT1 and ferroportin in patients with IDA, regardless of the presence of CD. Other studies have shown a greater expression of ferritin in celiac patients with IDA respect than in those without CD [57].

Toulon et al. showed that the DMT1-IVS4 + 44-AA polymorphism increased the risk of developing anemia, regardless of the degree of atrophy, approximately four times in 387 celiac children compared to 164 control children. In fact, the A allele seems to result in the reduced overexpression of DMT1 necessary in the case of iron deficiency [23].

Other studies investigated if mutations of hemochromatosis genes (HFE), that increase intestinal absorption of iron, could protect CD from IDA. Barisani et al. did not find a protective effect of HFE mutations investigating 203 patients with CD [56]. De Falco et al., on the other hand, demonstrated, with a comparative study of the HFE variants C282Y, H63D and TMPRSS6 in 505 patients with CD at diagnosis and after one year of GFD versus 539 control subjects, how HFE mutations protect celiac patients from the onset of IDA [20]. It was also co-confirmed as TMPRSS6, which modulates the action of epicidin and regulates the oral response of iron [20,55].

In confirmation of these data, Elli et al. showed a higher prevalence and significantly higher TMPRSS6 mutations in celiac patients than in controls, without however demonstrating differences between IDA and non-IDA in patients with CD [58].

Chronic disease anemia (ACD) occurs as a result of an abnormal activation of the immune system following the release of inflammatory cytokines. Among these, it has been shown that, in particular, the interferon-gamma (IFN-y), the interleukin-6 (IL-6) and the tumor necrosis factor (TNF) modulate the synthesis of the iron regulating hormone hepcidin. The latter determines the degradation of ferroportin and inhibits the release of iron by macrophages and enterocytes, thus modifying the reallocation outside the serum [1,2,3].

Systemic inflammation, subsequent to the increase in blood levels of inflammatory proteins, is a rare event in patients with CD. However, gliadin can favor the activation of mononuclear cells, located in the intestinal lamina propria mucosa, with subsequent local overproduction of proinflammatory cytokines such as IFN-y and IL-6 which favor the onset of ACD [57]. Bergamaschi et al. demonstrated a prevalence of ACD in 17% of CD patients [14]. The data were confirmed by Harper et al., although it was pointed out that patients with CD and ACD do not show signs of systemic inflammation [51]. The percentage instead reported by Berry et al. fluctuates around 3.9% [15].

GFD favors the improvement of intestinal atrophy but also induces a reduction in inflammation with therefore progressive correction of anemia. The mechanism is therefore twofold: increased iron absorption and reduced effects of various inflammatory mediators on iron homeostasis and erythropoiesis.

Therefore, it is evident that the anemia found in subjects with CD has a multifactorial pathogenesis: the most common form is secondary to iron deficiency, especially in patients with more extensive and severe mucosal atrophy, but malabsorption of folate and vitamin B12, a loss of blood or ACD [57].

In the study of Berry, 93% of patients with CD showed anemia, with IDA being the most common cause (81.5%). Other causes included folate deficiency (10.7%), vitamin B12 deficiency (13.6%), mixed nutritional deficiency (16.5%), and ACD (3.9%) [15].

Finally, blood loss due to inflammatory lesions of intestinal mucosa may contribute to IDA in celiac patients, as confirmed in the results provided by Martin Masotet al. [57] and Elli et al. [58].

## 3. Diagnostic Workup for IDA Diagnosis in CD Patients

We speak of anemia when the blood concentration of hemoglobin (Hb) is less than <2 SD of the normal values, which vary according to age, sex, elevation, smoking habit and physiological conditions such as pregnancy [1,2,3].

IDA represents a form of hypochromic microcytic anemia. Generally, it is moderate (Hb 9–11 g/dl), rarely severe (<7 g/dl).

The peripheral blood smear shows hypochromic (pale) and microcytic erythrocytes of variable size and shape (aniso-poikilocytosis). A change in red blood cell distribution can be detected early, as assessed by red blood cell distribution width (RDW) and hemoglobin distribution (HDS). Iron deficiency is diagnosed by the presence of low serum iron levels (less than 50 µg/dl) and high serum transferrin levels. Another highly sensitive index for diagnosing IDA is the saturation index of tranferrin, which is less than 10–16%. Low ferritin levels represented an early and highly specific indicator of iron deficiency. However, the international criteria for defining depleted iron deposits vary with age: <12 μg/L for children under 5 years and equal to 15–20 μg/L for those over 5 years and adults.However, it is important to remember that since ferritin is an acute phase reactive protein, its cutoff for the diagnosis of IDA rises to 30–50 μg/L in case of infection or inflammation (correlated with the increase in protein levels C-reactive). More recently, other parameters have been included in the workup of IDA, but they are generally reserved for more complicated diagnosis. Values of Reticulocyte hemoglobin (Chr) less than 27.5 pg are considered very sensitive and specific for IDA (83% and 72% respectively). Other parameters useful for the diagnosis of IDA are an increase in the levels of protoporphyrin IX in red blood cells and zinc protoporphyrin (ZPP). Low levels of serum hepcidin, a new biomarker, can aid in diagnosis, sometimes becoming wearable in the more severe forms of IDA. Frequently, individuals with IDA may experience thrombocytosis (platelet count between 500,000 and 700,000 mm^3^) and in severe forms, a low degree of hemolysis can be observed due to the rigidity of the red blood cell membrane [1,2,3]. Table 2.

## 4. Prevention

Patients with CD must follow prevention measures for IDA as recommended also in the non-CD population according to the age, gender, lifestyle and other underlying causes in order to guarantee an adequate iron balance. Furthermore, as CD patients are at high risk for IDA they must be tested for this condition at the diagnosis of CD and during GFD [1,2,3].

## 5. Treatment

The key treatment for CD is GFD [59]. GFD alone may improve mild forms of IDA in patients with CD [59]. If, in addition to CD, other underlying causes of IDA exist, they must be addressed when possible, in order to improve iron balance [1,2,3].

Management of IDA is primary focused on iron stores repletion. To date, discussions remain open on the most suitable modality for iron supplementation, between oral or intravenous therapy [1,2,3].

The most commonly undertaken therapy for oral iron replacement is with ferrous sulphate (FS), as it is cheaper, easier to administer and presents no risk of life-threatening events. Unfortunately, treatment with FS is limited by gastrointestinal side effects such as abdominal pain, nausea, diarrhea, vomiting and constipation that interest approximately 50% of patients [1,2,3].

In children, the recommended iron dose is 2–6 mg/kg/day in terms of elemental iron. In adolescents and adults, it is 100–200 mg daily [1,2,3]. Numerous other forms of iron (bivalent or trivalent) have been commercially available for a few years. Generally, these types of ferrous are better tolerated than FS but are inferior to FS in effectiveness of iron replacement; furthermore, bivalent compounds result as more absorbed than trivalent ones [1,2,3]. Large variations were observed in mean non-heme iron absorption between studies, which depended on iron status (diet had a greater effect at low serum and plasma ferritin concentrations) and dietary enhancers and inhibitors. So far, it is well known that Vitamin C, hydrochloric acid, sorbitol, ethanol, lactic acid, tartaric acid as well as meat, poultry, fish, affect the increase in absorption while Vitamin E, phytates, (tea, coffee), polyphenols, calcium and dairy products, animal proteins (milk and eggs-albumin), micronutrients (zinc and copper) fiber, casein, legume proteins, calcium, magnesium carbonate and cigarettes contribute to the decrease in absorption. Interactions of iron with manganese, chromium and selenium are still under investigation [1,2,3].

Absorption of FS is limited in patients with active CD and unpredictable in patients on a GFD [54,55,56,57]. Furthermore, a poor tolerability is particularly frequent in patients with CD or with other inflammatory bowel diseases. A possible solution is the concomitant use of probiotic or prebiotic. Some studies have shown that the association of a probiotic such as *Lactobacillus plantarum 299v* and *Bifidobacteriumlactis HN019* could facilitate a better iron absorption, but the results are controversial [49,60]. Promising results are derived from a pilot clinical trial, which evaluated the synergistic effect of a prebiotic (oligofructose-enriched inulin) on iron homeostasis in children and adolescents with celiac disease treated with a gluten-free diet [61].

In non-celiac patients, the achievement of normal hemoglobin levels can be generally expected after 2–3 months of oral treatment that should be further continued for 2–4 months in order to fill the body stores [1,2,3,49]. In some cases of CD, it takes longer to improve intestinal lesions with GFD. On average, it is estimated that anemia occurs after about 6 to 12 months with GFD, but sometimes it takes up to 2 years [54,55,56,57].

Ferrous Bisglycinate Chelate (FBC) is a new product consisting of a chelated ferrous iron atom bonded to two glycine molecules, with covalent and coordinated bonds. Several studies have demonstrated the efficacy and safety in the treatment of IDA in both adults and children [62,63,64]. Our study, through the use of an oral iron absorption test (OIAT) with FBC, has an excellent absorption profile of this product in children with newly diagnosed CD and in patients with GFD, without showing side effects [65,66].

A new FBC compound, called Feralgine^®^, has recently been developed to improve the bioavailability and tolerability profile. It is a patented compound, with a one-to-one ratio between FBC and sodium alginate using spray drying technologies [67].

Feralgine^®^, as well as FBC, is effective at a dosage of 30–40% compared to FS. Vernero et al. evidenced that the compound is well absorbed and tolerated in patients with inflammatory bowel disease [68]. Two of our recent studies conducted with OIAT in adult celiac patients confirmed the good level of absorption and tolerance in patients with anemia as well as in non-celiac subjects and in those with onset celiac disease [69,70].

The mechanisms by which this improved absorption occurs in CD subjects are not yet clear. Only the most accredited hypotheses are different; the first is that iron chelated with amino acids is better absorbed in the intestine than inorganic iron, perhaps through the use of different absorption processes. In fact, it has been shown that FBC increases the expression of the DMT1 transcript, PepT1 a heme iron transporter, hypothesizing that the latter favors the direct absorption of FBC as heme iron [71,72]. Furthermore, since the intestinal absorption of FBC occurs with the molecule intact, the iron complex is probably absorbed regardless of the presence of DMT1. However, further studies are needed to better clarify the absorption and bioavailability mechanisms of this product.

If oral iron is not tolerated, or not absorbed due to intestinal inflammation, then intravenous iron should be given. Intravenous (IV) administration of iron is the only alternative to oral administration, as intramuscular iron injections are no longer given due to the many side effects: excessive pain, abnormal skin discoloration and potential risk of developing injection site sarcoma (observed in model animals) [1,2,3,49]. Intravenous therapy is usually initiated when there is a severe form of IDA that requires rapid correction, forms in which a poor response to oral administration is demonstrated, or when there is difficulty in tolerability and adherence itself [1,2,3,49,73]. However, although there are numerous clinical studies on its efficacy, its application in daily management has been slow, partly due to the fear of possible adverse events, related to historical anaphylactic reactions associated with iron dextran formulations. Studies show that rates of mild reactions are ~1 in 200 and major reactions are ~1 in 200,000 or more [3,49,73].

Numerous various iron formulation for intravenous use have been developed in recent years. Their efficacy in the management of IDA is greater than possible adverse events, which can be easily managed if suitable measures are implemented to ensure early diagnosis and effective management of allergic reactions [3,49,73]. Low molecular weight iron dextran, introduced in the 1990s, although rarely causing serious reactions, is no longer used, due to the current availability of new formulations with an improved safety profile, such as: Fe-gluconate (Ferlixit^®^), Fe-sucrose (Venofer^®^) and Fe-carboxymaltose (Ferrinject^®^). New preparations such as Fe-isomaltoside (Monofer^®^) and Ferumoxytol (Ferraheme^®^) are currently being studied [3,49,73].

Recently, intravenous iron has been recommended in adults and in children with clinically active IBD, in case of intolerance to oral iron with interesting results [74,75]. Atthe moment there are no indications to treat IDA in CD patients with intravenous formulations. However, in case of severe anemia in patients with compromised conditions, it might be taken into considerations in order to rapidly correct the hematological picture as well as recommended in non-celiac patients.

Only when severe anemia is present, with hemoglobin values below 5–6 g/dl, which requires rapid correction such as in patients with cardiac dysfunction, red blood cell transfusions can be performed [76,77].

## 6. Conclusions

IDA represents the most frequent EIM of CD. Since patients with IDA, especially if a non-responsive form to treatment has a higher risk of having CD than the general population, screening with tissue transglutaminase antibodies is strongly recommended in these subjects.

If the mainstay of treatment for CD remains adherence to a GFD the management of IDA in CD is primary focused on iron stores repletion. For many patients, oral iron replacement with FS has long been the cornerstone treatment. However, treatment with FS can have some side effects, particularly on the gastrointestinal level and more frequent in patients with CD. Numerous other iron-based products (bivalent or trivalent) have been developed for some years, that are well tolerated than FS, but they may be less effective in correcting IDA, especially in people with CD. However, recent studies have shown that some of them, including Feralgine^®^, could show a good safety and efficacy profile even in celiac patients. Intravenous formulations are suggested only in patients with severe anemia and/or cardiac compromission who need rapid increase of hemoglobin levels.

## Figures and Tables

**Table 1 nutrients-13-01695-t001:** Studies on prevalence of IDA in celiac patients.

Authors	Country	No. Of Patients	% IDA	Year of the Study
ADULTS				
Koho, et al. [13]	Finland	8	25	1998
Bergamaschi et al. [14]	Italy	132	34	2008
Berry et al. [15]	India	103	81	2018
Binicier et al. [16]	Turkey	195	53	2020
Bottaro et al. [17]	Italy	315	46	1999
Abu Daya et al. [18]	USA	727	21	2013
Sansotta et al. [19]	USA	327	48	2018
De Falco et al. [20]	Italy	505	45	2018
Akbari et al. [21]	Iran	27	52	2006
Kockar et al. [22]	India	434	84	2012
CHILDREN				
Bottaro et al. [17]	Italy	485	35	1999
Sansotta et al. [19]	USA	227	12	2018
Tolone et al. [23]	Italy	385	35	2017
Carroccio et al. [24]	Italy	130	70	1998
Kullogu et al. [25]	Turkey	109	82	2009
Sanseviero et al. [26]	Italy	518	22	2016

**Table 2 nutrients-13-01695-t002:** Parameters for IDA diagnosis.

(a) Red cell parameters values for diagnosis of IDA
⮚ Reduction of Hb, RBCs and hematocrit < 2 SD of normal values according to age and gender. For WHO in adult, anemia is defined as hemoglobin < 13 g/dL in men and < 12 g/dL in non-pregnant women. In children, reference values are lower and differ according to age. ⮚ Reduction of MCV, MCH and MCHC ⮚ Hypochromic cells with a tendency to microcytosis ⮚ Increase of RDW > 15% ⮚ Reduction of CHr < 27.5 pg
(b) Biochemical parameters values for diagnosis of IDA
⮚ Reduction of serum iron < 30mg/dL; increase of total serum transferrin or of TIBC> 350 mg/dL; reduction of IS < 16%; ⮚ Reduction of serum ferritin < 10–20 ng/mL if PRC is normal. A ferritin threshold value of <45 ng/mL has a sensitivity for iron deficiency of 85% with a specificity of 92%. In contrast, a ferritin value of < 15 ng/mL has a sensitivity of only 59% and specificity of 99%. A ferritin threshold value of < 45 ng/mL is believed to maximize sensitivity for the diagnosis of IDA with an acceptable number of false-positive diagnoses.
(c) Other parameters evaluable for diagnosis of IDA
⮚ Increase of sTfR to a 10–14 mg/L ⮚ Reduction of reticulocyte (incostant) ⮚ Increase of zincoprotoporftina> 60–80 µmol/mol-heme ⮚ Increase of Free Erytrhrocyte Protoporphyrin (FEP) > 10 mg/dL ⮚ Increase of platelets count (incostant) between 600.000–1000.000 mmc. ⮚ Rarely modest hemolysis

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
