# Peer review of "Iron Deficiency Anemia in Celiac Disease"

_nutrients, 2021, doi:10.3390/nu13051695_

Round 1
Reviewer 1 Report
The work is interesting and is a review of current knowledge on iron and iron deficiency diseases but requires some minor additions
One of the topics discussed in the presented study is iron supplementation, in particular in children and the recommended iron dose. There is no recommendation for the form of supplements for children, no rules for taking them so that the use of iron is the best, and no description of adverse reactions during supplementation. It would also be interesting to present an overview of iron-containing preparations that can be safely used by people with celiac disease.
lines 245-248 - what probiotic strains can improve the absorption and absorption of iron
Are there studies that show differences in the effectiveness of specific chemical forms of iron compounds (inorganic and organic forms, complexes, chelates) in people suffering from celiac disease with inflammation and damage to the intestinal villi and in people with normal intestinal condition?
The authors write about iron deficiency anemia and the methods of its treatment do not mention the diet and food products that are the source of iron (heme and non-heme) and the principles of composing a diet that promotes better absorption of iron (the presence of vitamin C, animal protein) or those ingredients that reduce digestibility iron from both the diet and supplements (phytates, oxalates, fiber, other divalent ions). It is necessary to supplement the section Treatment with a brief description of the effects of food ingredients and the gluten-free food itself.
Author Response
Dear reviewers/editorial board
The authors of the manuscript titled: “Iron Deficiency Anemia in Celiac Disease” thank for the attention you are paying on our work. We made all the changes requested, hoping to have made the work appropriate for publication in the journal.
In particular:
Reviewer 1
Point 1: One of the topics discussed in the presented study is iron supplementation, in particular in children and the recommended iron dose. There is no recommendation for the form of supplements for children, no rules for taking them so that the use of iron is the best, and no description of adverse reactions during supplementation. It would also be interesting to present an overview of iron-containing preparations that can be safely used by people with celiac disease.
Response 1: As expressed in the text the current recommendations on the dosage to be performed in pediatric age is 2-6 mg/kg/day in term of elemental iron. In adolescents and adults it is 100-200 mg daily [1-3]. We reported the main adverse reactions reported during iron therapy: “Unfortunately, treatment with FS is limited by gastrointestinal side effects as abdominal pain, nausea, diarrhea, vomiting and constipation, that interest approximately 50% of patients [1-3].
To the best of our knowledge there are not iron containing preparations that can be absolute safetely used by people with celiac disease. We reported the most important data from some pilot study with Ferrous Bisglycinate Chelate and Ferrous Bisglycinate Chelate alginate.
Point 2: lines 245-248 - what probiotic strains can improve the absorption and absorption of iron
Response 2: We have included in the manuscript the probiotic strains that can improve absorption of iron (Lactobacillus plantarum 299v and Bifidobacteriumlactis HN019)
Point 3: Are there studies that show differences in the effectiveness of specific chemical forms of iron compounds (inorganic and organic forms, complexes, chelates) in people suffering from celiac disease with inflammation and damage to the intestinal villi and in people with normal intestinal condition?
Response 3: Concerning the people suffering from celiac disease there are not comparing studies of different iron compounds. Our non-randomized pilot studies showed that Ferrous Bisglycinate Chelate and Ferrous Bisglycinate Chelate alginate are well absorbed in celiac patients and well tolerated. In people with normal intestinal conditions some studies compared different iron compounds to iron sulphate that remains the gold standard.
Point 4: The authors write about iron deficiency anemia and the methods of its treatment do not mention the diet and food products that are the source of iron (heme and non-heme) and the principles of composing a diet that promotes better absorption of iron (the presence of vitamin C, animal protein) or those ingredients that reduce digestibility iron from both the diet and supplements (phytates, oxalates, fiber, other divalent ions). It is necessary to supplement the section Treatment with a brief description of the effects of food ingredients and the gluten-free food itself.
Response 4: It is well documented that Gluten-free diet improves iron-deficiency anaemia in patients with celiac disease, and we stressed this concept in the manuscript. We also added in the work, as requested, the list of the main foods that can interfere with the absorption of iron
We have modified som references to better understand various concepts requested by the reviewer
Best regards
Reviewer 2 Report
This is a very interesting review about a relevant association on CE, because the IDA is very common in these patients
The authors have comments about the pathophysiology of the intestinal absorption and the main role that plays in their production
The laboratory findings are several and the role that they play is very well explained
The treatment is based in the implantation of a strict GFD as the main support combined in some cases with drug iron replacement by oral route an by intravenous in few occasions
Its prognosis is generally excellent and full recovery is the rule , but is necessary to wait between 6-24 months in special cases
The review is very well planned and clearly written
Q1 : Title
Is short, good and very expressive
Q2 : Abstract and Keywords
Is well summarized and complete
The keywords are well selected and enough
Q3 : 1. Introduction
On Table 1 is clearly exposed the great variability of the prevalence of IDA in CD patients including studies performed in the same country, both in adults and also in children
Surprisingly in the Paez study on CD pts. , IDA was found in 69.4% in pts without gastrointestinal symptoms, compared to 11.5% in the asymptomatic group
Smukalla found that hematologists rarely screen for celiac disease in the initial workup of IDA patients. According to these authors, this attitude could be a determinant of the underdiagnosis of CD in the United States
After about 24 months of a GFD an improvement in IDA can be observed showed a persistence of anemia after one year of GFD even in the presence of histological normalization of the duodenal mucosa.
Two other large series subsequently confirmed the hypothesis of correlation between severity of anemia and severity of intestinal atrophy . Similar results have been obtained in children
A normalization of the hemoglobin value was demonstrated after about 6-12 months from the beginning of the GFD alone, following the restoration of the integrity of the intestinal mucosa.
There are several factors that could explain the reason for the persistence of IDA after the adoption of a GFD and oral iron supplementation: non-adherence to a GFD, the presence of ultrastructural and/or molecular alterations of the enterocytes despite the reformation of the integrity of the duodenal mucosa or the presence of specific genetic factors.
Q4.2. Pathogenesis
GFD favors the improvement of intestinal atrophy but also induces a reduction in inflammation with therefore progressive correction of anemia. The mechanism is therefore twofold: increased iron absorption and reduced effects of various inflammatory mediators on iron homeostasis and erythropoiesis.
Therefore, it is evident that the anemia found in subjects with CD has a multifactorial pathogenesis: the most common form is secondary to iron deficiency, especially in patients with more extensive and severe mucosal atrophy, but malabsorption of folate and vitamin B12, a loss of blood or chronic disease anemia
In the study of Berry, 93% of patients with CD showed anemia, with IDA being the most common cause (81.5%). Other causes included folate deficiency (10.7%), vitamin B12 deficiency (13.6%), mixed nutritional deficiency (16.5%), and ACD (3.9%)
Finally, blood loss due to inflammatory lesions of intestinal mucosa may contribute to IDA in celiac patients, as confirming in the results provided by several published studies
Q4.3. Diagnostic workup for IDA diagnosis in CD patients
IDA represents a form of hypochromic microcytic anemia. Generally it is moderate (Hb 9-11 g/dl), rarely severe (<7 g/dl).
The peripheral blood smear shows hypochromic (pale) and microcytic erythrocytes of variable size and shape (aniso-poikilocytosis). A change in red blood cell distribution can be detected early, as assessed by red blood cell distribution width (RDW) and hemoglobin distribution (HDS). Iron deficiency is diagnosed by the presence of low serum iron levels (less than 50 μg / dl) and high serum transferrin levels.
Low ferritin levels represented an early and highly specific indicator of iron deficiency.
However, it is important to remember that since ferritin is an acute phase reactive protein, its cut-off for the diagnosis of IDA rises to 30-50 μg/L in case of infection or inflammation (correlated with the increase in protein levels C-reactive).
Frequently, individuals with IDA may experience thrombocytosis (platelet count between 500,000 and 700,000 mm3) and in severe forms, a low degree of hemolysis can be observed due to the rigidity of the red blood cell membrane.
Q4.4. Prevention
Patients with CD must follow prevention measures for IDA as recommended also in non-CD population according to the age, gender, lifestyle and other underlying causes in order to guarantee an adequate iron balance. Furthermore as CD patients are at high risk for IDA they must be tested for this condition at the diagnosis of CD and during GFD
Q5. Treatment
Management of IDA is primary focused on iron stores repletion.
The most commonly undertaken therapy for oral iron replacement is with ferrous sulphate (FS), as it is cheaper, easier to administer and presents no risk of life-threatening events. Unfortunately, treatment with FS is limited by gastrointestinal side effects that interest approximately 50% of patients
Feralgine®, as well as FBC, is effective at a dosage of 30-40% compared to FS. Vernero et al. evidenced as the compound is well absorbed and tolerated in patients with in- flammatory bowel disease
If oral iron is not tolerated, or not absorbed due to intestinal inflammation, then intravenous iron should be given.
At the moment there are no indications to treat IDA in CD patients with intravenous formulations. However, in case of severe anemia in patients with compromised conditions it might be taken into considerations in order to rapidly correct the hematological picture as well as recommended in non celiac patients.
Only when severe anemia is present, with hemoglobin values below 5-6 g/dl, which requires rapid correction such as in patients with cardiac dysfunction, red blood cell transfusions can be performed
Q6. Conclusions
IDA represents the most frequent EIM of CD. Since patients with IDA, especially if a non-responsive form to treatment has a higher risk of having CD than the general population, screening with tissue transglutaminase antibodies is strongly recommended in these subjects.
If the mainstay of treatment for CD remains adherence to a GFD the management of IDA in CD is primary focused on iron stores repletion.
For many patients oral iron replacement with FS has long been the cornerstone treatment. However, treatment with FS can have some side effects, particularly on the gastrointestinal level and more frequent in patients with CD
However, recent studies have shown that some of them, including Feralgine®, could show a good safety and efficacy profile even in celiac patients. Intravenous formulations are suggested only in patients with severe anemia and /or cardiac compromission who need rapid increase of hemoglobin levels
Q7. References
They are well selected and good enough in quality and in total number
Author Response

(The authors gave the same response as above.)
